# Vine Copula-Based Data Generation for Machine Learning With an Application to Industrial Processes

**Jean-Thomas Sexton**
Department of Computer Science and Software Engineering
Université Laval
Québec, QC, Canada G1V 0A6
jtsex@ulaval.ca

**Michael Morin**
Department of Operations and Decision Systems
Université Laval
Québec, QC, Canada G1V 0A6
michael.morin@osd.ulaval.ca

**Jonathan Gaudreault**
Department of Computer Science and Software Engineering
Université Laval
Québec, QC, Canada G1V 0A6
jonathan.gaudreault@ift.ulaval.ca

## Abstract

Synthetic data generation of industrial processes exhibiting non-stationarity and complex, non-linear dependencies between their inputs and outputs is a challenging task. We argue that vine copula models are particularly well suited for this problem and present a method combining limited available data and expert knowledge in order to generate synthetic data by conditionally sampling from a C-Vine, a type of vine copula. We demonstrate our approach by generating synthetic data for a high-speed, sophisticated lumber finishing machine called a wood planer.

## 1 Introduction

Machine learning is becoming an indispensable tool in order to provide proactive control of industrial processes [3]. However, due to current trends towards worker shortages, experienced operators with the knowledge necessary to control complex industrial processes are increasingly difficult to find in many industries. Therefore, as was noted in [8] in the context of building a machine learning-based control loop for a wood planer, complex machinery with large numbers of controllable settings is often operated using only a small subset of the available settings which can not only negatively affect the quality of the production output, but also that of the data collected for machine learning-based control. In addition, when a number of inputs are processed by a machine in a given production period, e.g., a *batch* of lumbers in the case of a wood planer, variability in the nature of the inputs (e.g., unrecorded mechanical adjustments and other factors such as equipment wear and tear) lead to non-stationarity which, in turn, leads to controllable settings having different impacts on the output in different batches which information might not be available in the data. For instance, in the case of a wood planer, a frozen lumber will necessitate more aggressive settings than a non-frozen lumber.

NeurIPS 2022 Workshop on Synthetic Data for Empowering ML Research.

To address these challenges, it would be useful to be able to train a machine learning model on a realistic superset of the possible configurations of dependencies between inputs and outputs that were observed in the real-world data. We make the hypothesis that a model able to identify (or approximate) the current context in this more difficult, but realistic, setting would be able to perform well in the simpler setting encountered in the real world. However, generating realistic synthetic data for industrial processes exhibiting a large number of variables, complex non-linear dependencies and a non-stationary behavior is a challenge, especially when real-world data encompassing all possible usage scenarios is scarce. We argue that *vine copulas* models [1] are particularly well suited for this task.

Using vine copulas for synthetic data generation has been explored notably in [10] and [7]. However, these approaches require more data than is available in our context. We propose a hybrid approach using expert knowledge and leveraging available real-world data to create vine copulas-based models able to generate synthetic data to build the aforementioned superset of dependencies between inputs and outputs. We illustrate our approach with an application to data generation for machine learning in the context of industrial wood planing.

The paper is organized as follows. In Section 2, we review vine copulas and highlight properties useful to our method. Then, in Sections 3 and 4, we detail the proposed approach and illustrate it with a practical example involving a wood planer. We conclude in Section 5.

## 2 Vine Copulas

We will first review *copulas*, the basic building block of vine copulas. Copulas allow the coupling of arbitrary univariate marginal distributions into a multivariate joint distribution. Using the same marginal distributions, we can use copulas to construct multivariate joint distributions with different dependencies structures. This is best illustrated with Sklar's theorem [9], included here as Theorem 1.

**Theorem 1** *For a random vector $\boldsymbol{X} \in \mathbb{R}^d$ where $d > 1$ with cumulative distribution function $\boldsymbol{F}$ and univariate marginal cumulative distribution functions $F_1, \ldots, F_d$, there exists a copula $C$ such that*

$$F(x_1, \ldots, x_n) = C(F_1(x_1), \ldots, F_d(x_d)).$$

*If $\boldsymbol{X}$ is continuous, then $C$ is unique.*

The implication of Theorem 1 is that any multivariate distribution can be modeled with two separate parts: its marginal distributions and a copula. This particularly useful in our case since the marginal distributions of a machine's settings are often simple to model using their empirical distributions or expert knowledge, leaving us with the task of modeling the dependency structure using a copula. However, it is difficult to construct copulas in dimensions higher than two and most copulas can only model a bivariate distribution [6].

This limitation is what led to the development of vine copulas for modeling higher dimensional distributions. A vine copula is a hierarchical model where a multivariate distribution function is decomposed with bivariate building blocks. Although such decompositions are not unique, they can be expressed as a graphical model composed of a sequence of $d - 1$ nested trees. We will use a particular subset of vine copulas named *C-Vines* where one node in each tree is maximally connected. This allows for greater explainability and thus allows us to integrate expert knowledge. In addition, C-Vines can be conditionally sampled in order to produce realizations from a certain number of predetermined (or simulated) variables [2]. We refer to [4] and [6] for further details on vine copulas. Examples of vines are available in Section 4.

## 3 Synthetic Data Generation with Vine Copulas

Conceptually, our approach consists of creating a C-Vine structure from expert knowledge by taking into account the physical relationships between the settings and the output. As building blocks, we only use bivariate copulas which can be parameterized using a single value (Kendall's $\tau$, a scale-free measure of rank correlation, where -1 indicates full negative correlation, 0 no correlation and 1 full positive correlation). The procedure for choosing this interval is described in Section 4. When generating synthetic data for a new batch, we randomly select a $\tau$ value for each pair, thus modeling

the fact that the same settings can have different effects in different contexts. In other words, we create a general dependency structure based on the physical characteristics of the process and randomly fix the strength of the dependencies when generating new synthetic data.

The features of an industrial process can be split into two categories: those that can be controlled such as the settings of a machine and those that cannot such as features inherent to the input or features describing the state of the process (e.g., the temperature of a machine). Our C-Vine structure is constructed by selecting the output first, then the non-controllable settings and then the controllable settings. This allows us to generate synthetic data for the output and non-controllable settings by conditionally sampling the C-Vine based on the values of the controllable setting.

In order to simulate adjustments made by operators to the controllable settings, we use distinct Poisson processes, one per controllable setting, where a new value for these variables is randomly sampled at every arrival time. The average inter-arrival time is calculated from the desired number of adjustments in the batch and the desired number of data points to be generated. In addition, time can be taken into account by using it as our last variable which is simply incremented from one input to the next. This method allows us to generate a large number of synthetic batches by simulating adjustments to the controllable settings using Poisson processes for the desired length of the production run which we will then use to generate the associated values of non-controllable settings and outputs by using our C-Vine.

The dependency structure modeled by the C-Vines remains determined by the physical nature of the industrial process, as understood by an expert, but the strength of the dependencies between pairs of variables vary from one run to the next more than it would in the real world. This enables us to test the robustness of different machine learning models in a more difficult setting then what is possible in a context where most controllable settings remain untouched.

## 4    Application to Synthetic Data Generation for a Wood Planer

Wood planers deal with the finishing of lumber products. They sequentially process lumber pieces in order to adjust their dimensions before being sold on the market. Being sophisticated and fast machines, they are difficult to operate and the available data shows complex, non-linear dependencies which vary from one production run to the next between controllable settings and the output, which in this case is the thickness of the processed lumber pieces.

In what follows, we use the data for 118,792 boards split in 83 production runs (batches) gathered during a 13-day operation period in a sawmill in Québec, Canada. Due to the high monetary cost of mistakes and the difficulty of finding experienced operators, most controllable settings remain untouched by operators, making it an appropriate case study for our method.

We consider ten features identified by experts as being the most important consisting of the output value, one non-adjustable setting, seven adjustable settings and time. Table 1 summarizes the features.

Table 1: Features used for generating synthetic wood planer data

| Feature | Type |
| --- | --- |
| Thickness of the processed board | output |
| Temperature of the machine | non-controllable setting |
| Offset | controllable setting |
| Position of the knives | controllable setting |
| Speed of the roller | controllable setting |
| Pressure of the roller | controllable setting |
| Position of the bedplate | controllable setting |
| Position of the upper cylinder | controllable setting |
| Position of the lower cylinder | controllable setting |
| Timestep | Time |

When constructing our C-Vine, we work backwards through these variables starting with the timestep as the maximally connected node in the first tree and ending with the temperature as the maximally connected node in the tenth tree. We will detail the construction of the first two trees. The full

structure with Kendall's $\tau$ intervals is included in the Appendix. We refer to [6] for a complete description of the different bivariate copulas we used.

For the first tree, we have to assign a copula between every variable and the timestep. Since experts indicated that the only variable affected directly by the passing of time is the temperature, which always increases over time, we thus use the independence copula (I) for every variable except temperature. Available data shows that Frank's copula is an appropriate choice, with Kendall's $\tau$ being between 0.3 and 0.5 in 100% of the batches. Since we want to model a realistic superset of the dependencies, we use expert knowledge to expend our interval of possible Kendall's $\tau$ for the dependency between the temperature variable and the timestep to $[0.2, 0.6]$. This first tree is illustrated in Figure 1.

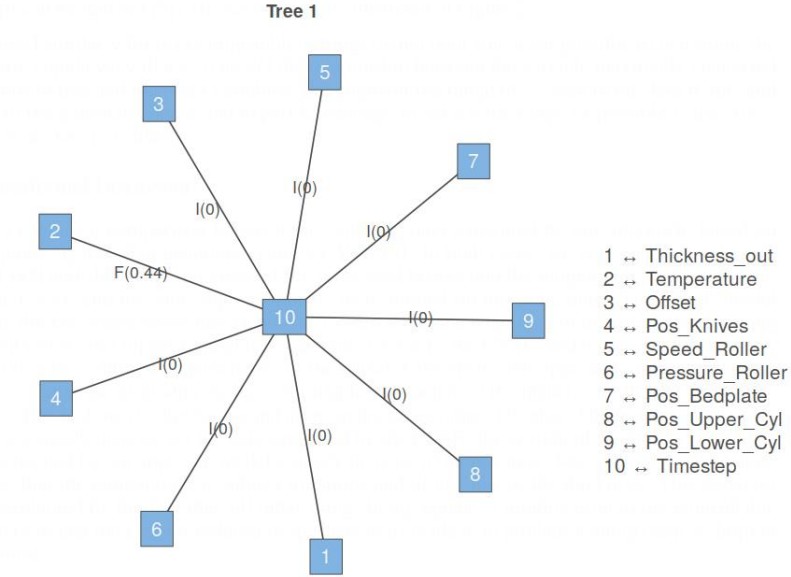

Figure 1: First tree of the C-Vine, every variable is coupled with the Timestep using the independence copula (I(0)) except for the temperature of the machine where the copula used is the Frank's copula (F) with a $\tau$ value randomly selected in $[0.2, 0.6]$ (here 0.44)

For the second tree, we have to assign a copula between every remaining variable and the position of the lower cylinder. Since all adjustable settings are physically independent, we will systematically choose the independence copula to model their relationships. Unfortunately, no data is available with regards to the effect that moving the position of the lower cylinder has on the output or the temperature. However, we know from the physical characteristics of the machine that raising the lower cylinder will lead to thinner boards and higher temperatures while lowering it will lead to thicker boards and lower temperatures. We thus use a Gaussian copula (a sensible choice in the absence of data) to model the dependency between the position of the lower cylinder and the thickness of the processed boards with $[-0.8, -0.05]$ as a range of Kendall's $\tau$s to express the inverse relationship between those two variables. Similarly, we use a Gaussian copula to model the dependency between the position of the lower cylinder and the temperature with $[0.05, 0.8]$ as the interval of plausible Kendall's $\tau$. We note that we use the common simplifying assumption that the conditioning variable(s) from the previous tree (here the timestep) can be ignored [5]. The second tree is illustrated in Figure 2.

We proceed similarly for each adjustable setting, using data whenever possible to determine the parametric copula we will use to model the relationship between the variable maximally connected in the current tree and the other variables. We augment the range of $\tau$s seen in the data, if any, by using expert knowledge. We default to the Gaussian copula and, again, expert knowledge to set a wide range of plausible values for $\tau$ whenever no or little data is available.

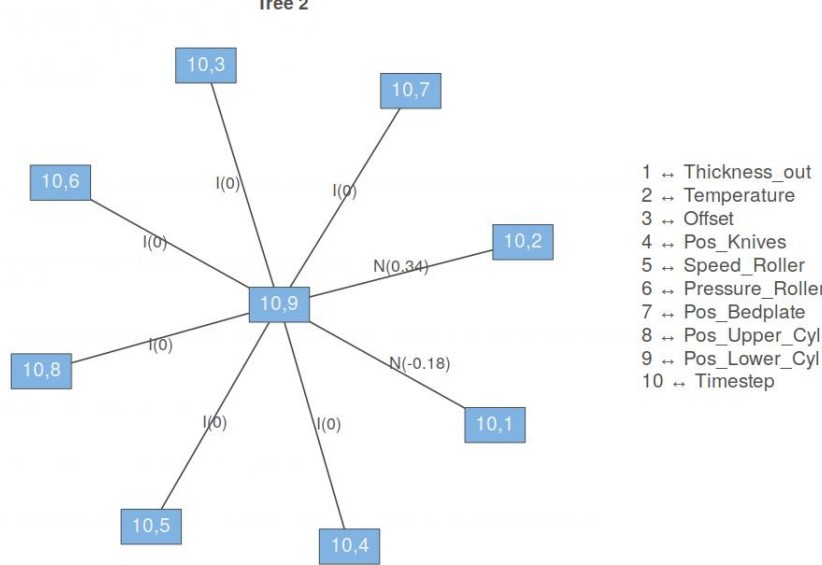

Figure 2: Second tree of the C-Vine, every variable is coupled with the position of the lower cylinder (ignoring the conditioning variable) using the independence copula (I(0)) except for the thickness of the processed board and the temperature of the machine. Here the copula used in both cases is the Gaussian copula (N) with a randomly selected $\tau$ between $[-0.8, -0.05]$ and $[0.05, 0.8]$ respectively (here -0.18 and 0.34)

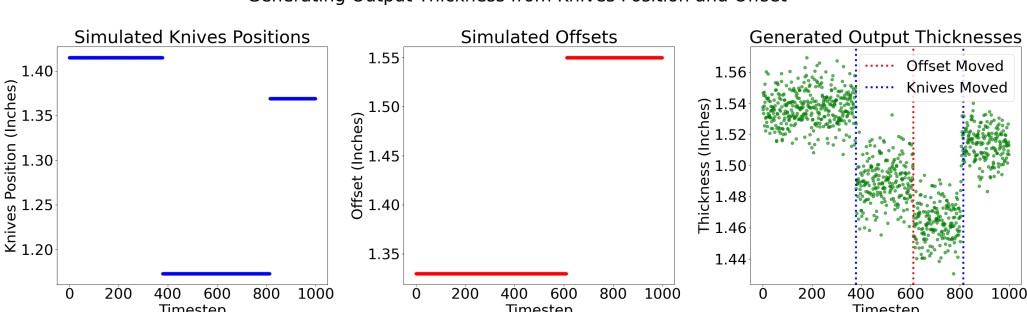

Figure 3: Simulated knives position using a Poisson process (left), simulated offsets (center) and generated output thicknesses for a production run of 1,000 boards from our C-Vine model

### 4.1 Results and Discussion

To illustrate the generated data, Figure 3 shows the effect of setting every adjustable parameter equal their respective median, except for the position of the knives and the offset. The adjustments made to those two adjustable parameters are generated using a Poisson process where the average inter-arrival time is set to 400 for the position of the knives and 600 for the offset. The value of the adjusted parameter is randomly chosen among the possible values of each parameter using a uniform distribution. The C-Vine used to generate the data is the same one that is described in the Appendix. We generated a production run of a batch of 1,000 boards and show the simulated knives positions, the simulated offsets and the generated output thicknesses.

The behavior of the generated output thicknesses was assessed by an expert as being realistic; as expected we can observe that the thickness of the outputs tends to be reduced when the position of the knives is lowered whereas it tends to be increased when they are raised (first and second vertical

dotted blue lines in Figure 3). Likewise, when the offset is raised, we can observe that the thickness of the outputs tends to be reduced (the vertical red dotted line in Figure 3).

## 5 Conclusion

Preliminary experiments have suggested our vine copulas can simulate/generate batches exhibiting a realistic behavior (as assessed by experts) w.r.t. combined planer input modifications even when very little data where adjustable settings are modified is available. The flexibility of being able to generate the controllable variables separately from the non-controllable variables and the output allows a wide range of variations in the adjustments made to controllable variables without having to change the underlying generative model. In addition, vine copulas give the ability to adjust the sensitivity of a given output to a given input simply by changing a single number whilst being able to capture complex, non-linear relationships which is greatly useful in practice for integrating expert knowledge and overcoming a lack of data. Future work will focus on assessing the performance of machine learning models trained with a large amount of synthetic production runs generated with our model, as well as rigorously assessing the quality of the generated data beyond expert opinion.

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

## Appendix

For completeness, we include every tree in the C-Vine in this section, including those already described in Section 4. Figures 4 to 12 show the structure of the trees in the C-Vine. Tables 2 to 10 show the copula families used and the interval of Kendall's $\tau$ used to model the relationship between the maximally connected node and the other variables for each tree in the C-Vine.

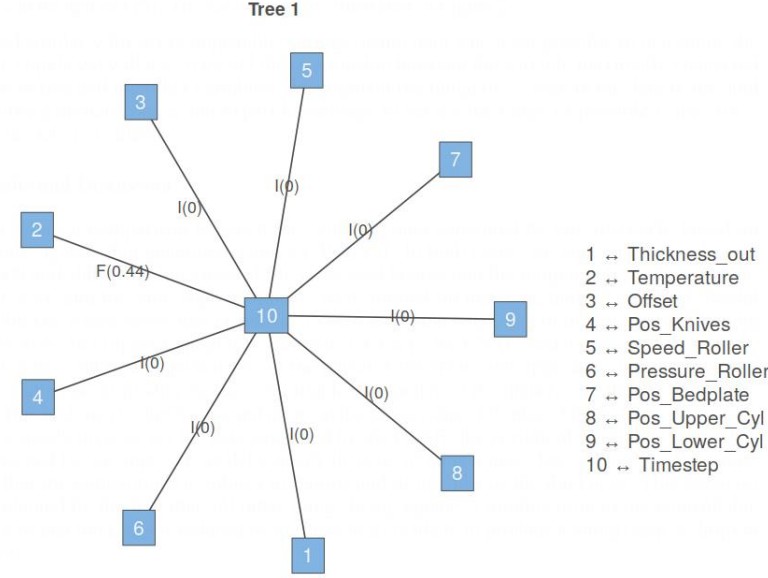

Figure 4: First tree of the C-Vine, every variable is coupled with the Timestep using the independence copula (I(0)) except for the temperature of the machine where the copula used is the Frank's copula (F) with a $\tau$ value randomly selected in $[0.2, 0.6]$ (here 0.44)

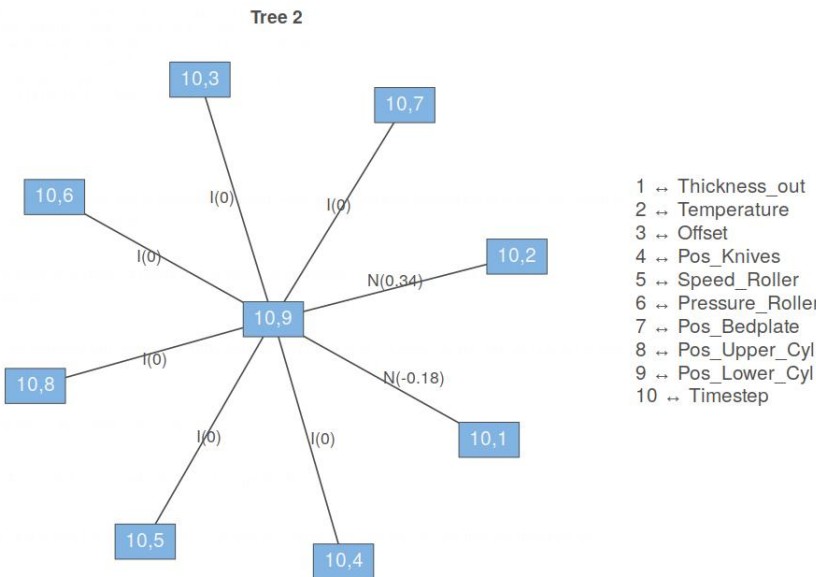

Figure 5: Second tree of the C-Vine, every variable is coupled with the position of the lower cylinder (ignoring the conditioning variable) using the independence copula (I(0)) except for the thickness of the processed board and the temperature of the machine. Here the copula used in both cases is the Gaussian copula (N) with a randomly selected $\tau$ between $[-0.8, -0.05]$ and $[0.05, 0.8]$ respectively (here -0.18 and 0.34)

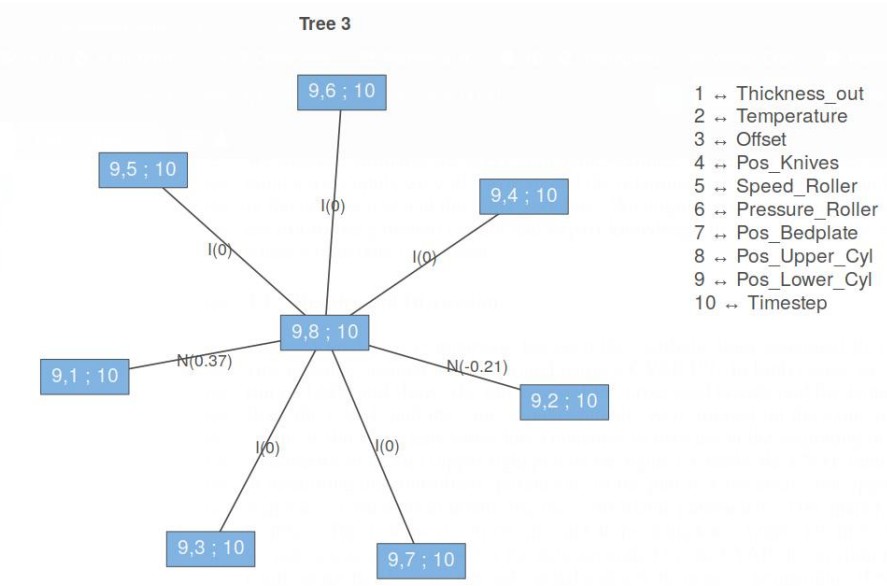

Figure 6: Third tree of the C-Vine, every variable is coupled with the position of the upper cylinder (ignoring the conditioning variables) using the independence copula (I(0)) except for the thickness of the processed board and the temperature of the machine where the copula used in both cases is the Gaussian copula (N) with a randomly selected $\tau$ between $[0.1, 0.6]$ and $[-0.6, -0.1]$ respectively (here 0.37 and -0.21)

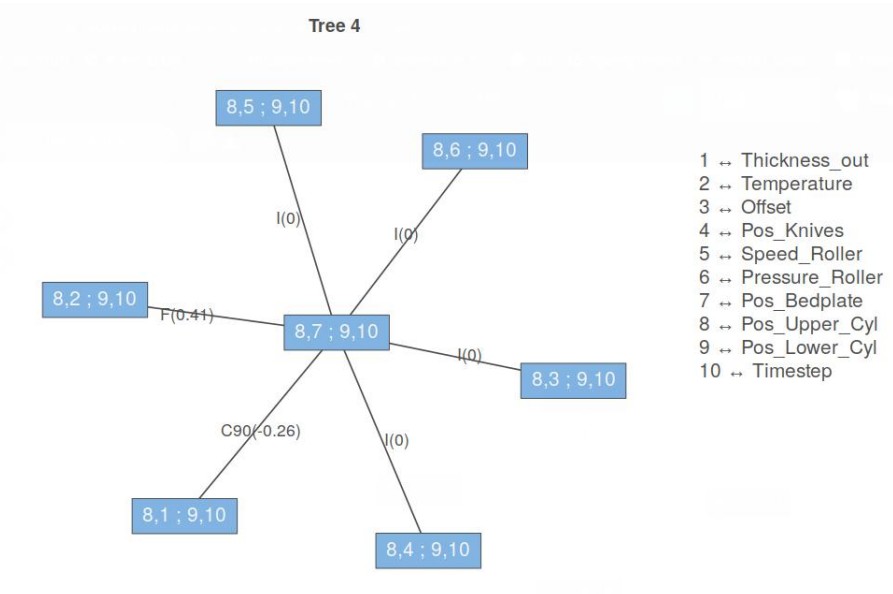

Figure 7: Fourth tree of the C-Vine, every variable is coupled with the position of the bedplate (ignoring the conditioning variables) using the independence copula (I(0)) except for the thickness of the processed board which uses a Clayton copula rotated by 90 degrees (C) and the temperature of the machine where the Frank copula (F) is used. In both cases we use a randomly selected $\tau$ between $[-0.5, -0.2]$ and $[0.2, 0.5]$ respectively (here -0.26 and 0.41)

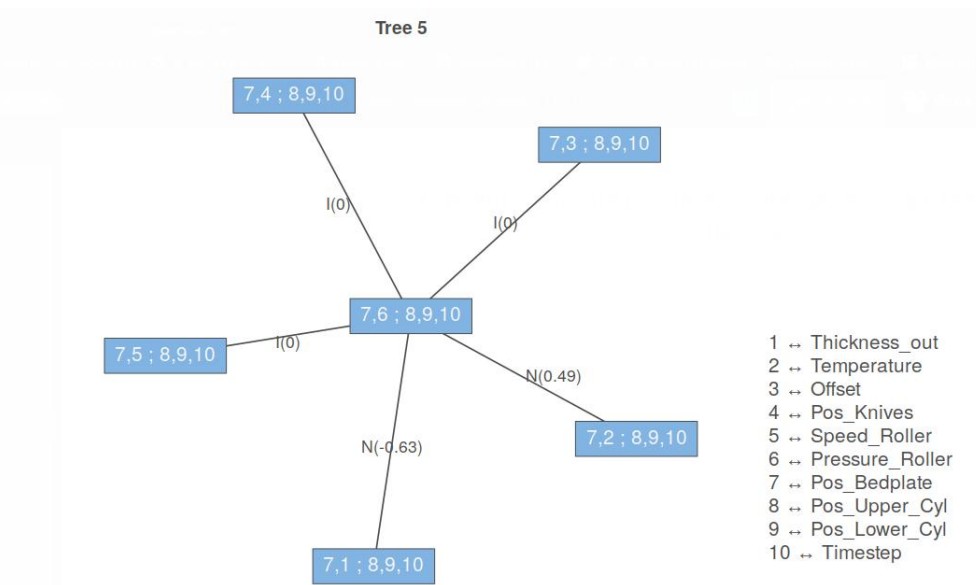

Figure 8: Fifth tree of the C-Vine, every variable is coupled with the pressure of the roller (ignoring the conditioning variables) using the independence copula (I(0)) except for the thickness of the processed board and the temperature of the machine where the copula used in both cases is the Gaussian copula (N) with a randomly selected $\tau$ between $[-0.8, -0.05]$ and $[0.05, 0.8]$ respectively (here -0.63 and 0.49)

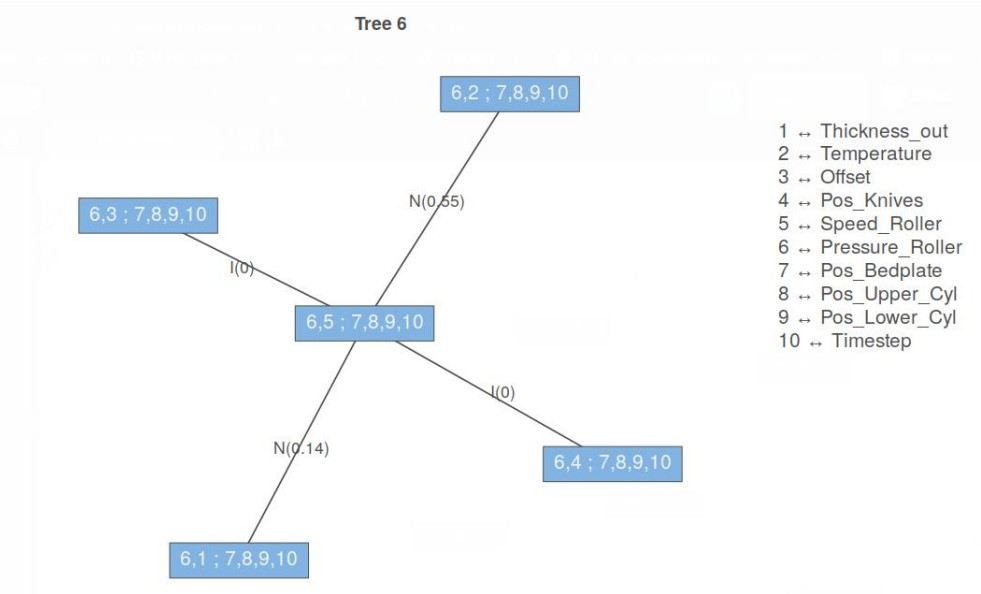

Figure 9: Sixth tree of the C-Vine, every variable is coupled with the speed of the roller (ignoring the conditioning variables) using the independence copula (I(0)) except for the thickness of the processed board and the temperature of the machine where the copula used in both cases is the Gaussian copula (N) with a randomly selected $\tau$ between $[0.05, 0.8]$ and $[0.05, 0.8]$ respectively (here 0.14 and 0.55)

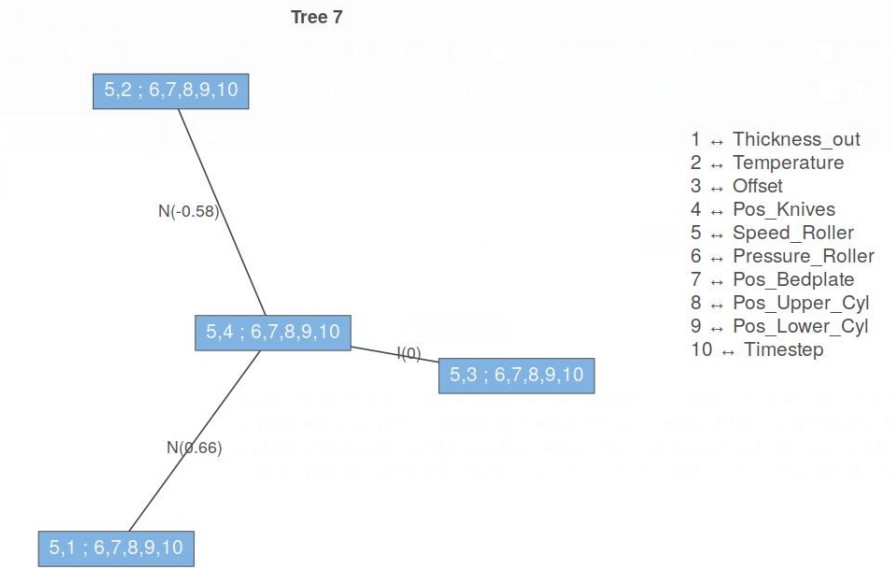

Figure 10: Seventh tree of the C-Vine, every variable is coupled with the position of the knives (ignoring the conditioning variables) using the independence copula (I(0)) except for the thickness of the processed board and the temperature of the machine where the copula used in both cases is the Gaussian copula (N) with a randomly selected $\tau$ between $[0.3, 0.9]$ and $[-0.9, -0.3]$ respectively (here 0.66 and -0.58)

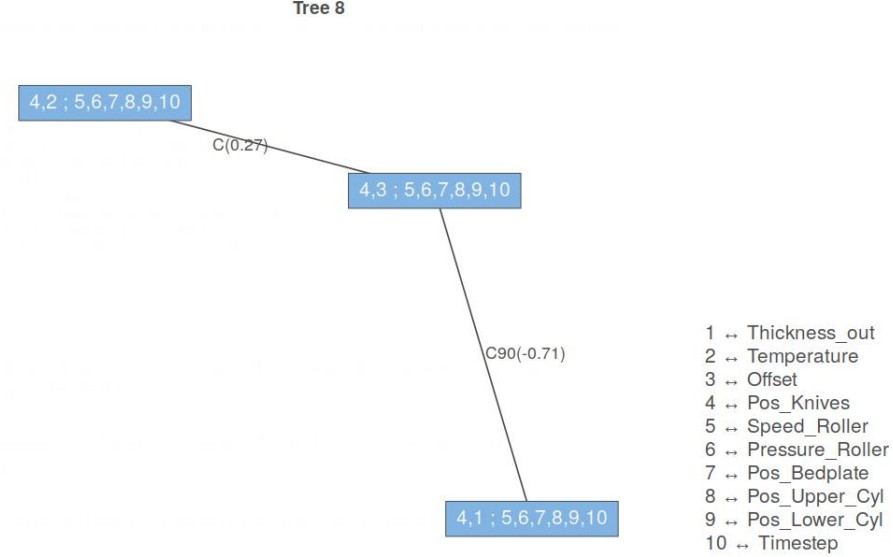

Figure 11: Eighth tree of the C-Vine, the last two variables are coupled with the offset (ignoring the conditioning variables), a Clayton copula rotated by 90 degrees (C90) is used for the thickness of the processed board and a Clayton copula (C) is used for the temperature of the machine where the copula used in both cases is the Gaussian copula (N) with a randomly selected $\tau$ between $[-0.9, -0.4]$ and $[0.1, 0.5]$ respectively (here -0.71 and 0.27)

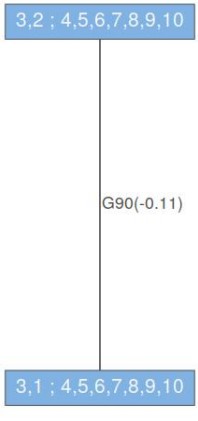

Tree 9

3,2 ; 4,5,6,7,8,9,10

G90(-0.11)

3,1 ; 4,5,6,7,8,9,10

1 ↔ Thickness_out
2 ↔ Temperature
3 ↔ Offset
4 ↔ Pos_Knives
5 ↔ Speed_Roller
6 ↔ Pressure_Roller
7 ↔ Pos_Bedplate
8 ↔ Pos_Upper_Cyl
9 ↔ Pos_Lower_Cyl
10 ↔ Timestep

Figure 12: Ninth tree of the C-Vine, the thickness of the processed board is linked to the temperature of the machine using a Gumbel copula rotated by 90 degrees with a randomly selected $\tau$ between $[-0.4, -0.05]$ (here -0.11)

Table 2: Copula used with each feature and range of Kendall's $\tau$ (if applicable) for the first tree

| Feature | Copula | Range of Kendall's $\tau$ |
| --- | --- | --- |
| Thickness of the processed board | Independence | N/A |
| Temperature of the machine | Frank | $[0.2, 0.6]$ |
| Offset | Independence | N/A |
| Position of the knives | Independence | N/A |
| Speed of the roller | Independence | N/A |
| Pressure of the roller | Independence | N/A |
| Position of the bedplate | Independence | N/A |
| Position of the upper cylinder | Independence | N/A |
| Position of the lower cylinder | Independence | N/A |

Table 3: Copula used with each feature and range of Kendall's $\tau$ (if applicable) for the second tree

| Feature | Copula | Range of Kendall's $\tau$ |
| --- | --- | --- |
| Thickness of the processed board | Gaussian | $[-0.8, -0.05]$ |
| Temperature of the machine | Gaussian | $[0.05, 0.8]$ |
| Offset | Independence | N/A |
| Position of the knives | Independence | N/A |
| Speed of the roller | Independence | N/A |
| Pressure of the roller | Independence | N/A |
| Position of the bedplate | Independence | N/A |
| Position of the upper cylinder | Independence | N/A |

Table 4: Copula used with each feature and range of Kendall's $\tau$ (if applicable) for the third tree

| Feature | Copula | Range of Kendall's $\tau$ |
|---|---|---|
| Thickness of the processed board | Gaussian | $[0.1, 0.6]$ |
| Temperature of the machine | Gaussian | $[-0.6, -0.1]$ |
| Offset | Independence | N/A |
| Position of the knives | Independence | N/A |
| Speed of the roller | Independence | N/A |
| Pressure of the roller | Independence | N/A |
| Position of the bedplate | Independence | N/A |

Table 5: Copula used with each feature and range of Kendall's $\tau$ (if applicable) for the fourth tree

| Feature | Copula | Range of Kendall's $\tau$ |
|---|---|---|
| Thickness of the processed board | Clayton90 | $[-0.5, -0.2]$ |
| Temperature of the machine | Clayton | $[0.2, 0.5]$ |
| Offset | Independence | N/A |
| Position of the knives | Independence | N/A |
| Speed of the roller | Independence | N/A |
| Pressure of the roller | Independence | N/A |

Table 6: Copula used with each feature and range of Kendall's $\tau$ (if applicable) for the fifth

| Feature | Copula | Range of Kendall's $\tau$ |
|---|---|---|
| Thickness of the processed board | Gaussian | $[-0.8, -0.05]$ |
| Temperature of the machine | Gaussian | $[0.05, 0.8]$ |
| Offset | Independence | N/A |
| Position of the knives | Independence | N/A |
| Speed of the roller | Independence | N/A |

Table 7: Copula used with each feature and range of Kendall's $\tau$ (if applicable) for the sixth

| Feature | Copula | Range of Kendall's $\tau$ |
|---|---|---|
| Thickness of the processed board | Gaussian | $[0.05, 0.8]$ |
| Temperature of the machine | Gaussian | $[0.05, 0.8]$ |
| Offset | Independence | N/A |
| Position of the knives | Independence | N/A |

Table 8: Copula used with each feature and range of Kendall's $\tau$ (if applicable) for the seventh tree

| Feature | Copula | Range of Kendall's $\tau$ |
|---|---|---|
| Thickness of the processed board | Gaussian | $[0.3, 0.9]$ |
| Temperature of the machine | Gaussian | $[-0.9, 0.3]$ |
| Offset | Independence | N/A |

Table 9: Copula used with each feature and range of Kendall's $\tau$ (if applicable) for the eighth tree

| Feature | Copula | Range of Kendall's $\tau$ |
|---|---|---|
| Thickness of the output | Gaussian | $[-0.9, -0.4]$ |
| Temperature of the machine | Clayton | $[0.1, 0.5]$ |

Table 10: Copula used with each feature and range of Kendall's $\tau$ (if applicable) for the ninth tree

| Feature | Copula | Range of Kendall's $\tau$ |
|---|---|---|
| Thickness of the processed board | Gumbel | $[-0.4, -0.05]$ |

