# OpenReview forum: "Vine Copula Based Data Generation for Machine Learning With an Application to Industrial Processes"
_NeurIPS.cc/2022/Workshop/SyntheticData4ML — Neurips 2022 SyntheticData4ML_

### Official Review · Reviewer_DhPR · 2022-10-17
**Interesting way of generating synthetic data but unclear way to measure usefulness**

**Rating:** 5
**Confidence:** 3

**Review:**

The paper proposes a vine copula model to generate synthetic data for an industrial process when the available training data is limited and expert knowledge is available.

Pros: The proposed method is simple and transparent making it easy to replicate and apply in different domains.

Cons: The evaluation method proposed relies on assessments by experts which can be variable and highly subjective. Also the paper doesn't show any usefulness of the generated data, i.e training downstream models on the generate synthetic data and evaluating their performance. This questions the effectiveness of the proposed approach which can't merely be estimated by assessments from domain experts.

---

### Official Review · Reviewer_YRDx · 2022-10-20

**Rating:** 7
**Confidence:** 3

**Review:**

This paper presents a novel approach for generating synthetic data using vine copulas. The vine copula models are created using a hybrid of real world data and expert knowledge. As such, this paper illustrates an interesting way to incorporate prior knowledge, i.e. the structural knowledge about the variable dependencies, into the generating process.

The synthetic data is evaluated by human experts. In future works, it would be interesting to show evaluation results in a more quantitative way.

---

### Meta-Review · Area_Chair_UM5X · 2022-10-20

**Recommendation:** Accept